# Molecular characterization and antibiotic resistance of *Acinetobacter baumannii* in cerebrospinal fluid and blood

Xiaohong Shi[1☉], Hong Wang[1☉], Xin Wang[2], Huaiqi Jing[2], Ran Duan[2], Shuai Qin[2], Dongyue Lv[2], Yufeng Fan[2], Zhenzhou Huang[2], Kyle Stirling[3,4], Lei Zhang[3,5], Jiazheng Wang[1]*

1 Department of Clinical Laboratory Shandong Provincial Qianfoshan Hospital, the First Affiliated Hospital of Shandong First Medical University, Jinan, Shandong, China, 2 State Key Laboratory for Infectious Disease Prevention and Control, National Institute for Communicable Disease Control and Prevention, Changping, Beijing, People's Republic of China, 3 Department of Biostatistics, School of Public Health, Cheeloo College of Medicine, Shandong University, Jinan, China, 4 Luddy School of Informatics, Computing and Engineering, Indiana University, Bloomington, IN, United States of America, 5 Institute for Medical Dataology, Cheeloo College of Medicine, Shandong University, Jinan, China

☉ These authors contributed equally to this work.
* wangjiazheng1210@163.com

**Data Availability Statement:** All relevant data are within the manuscript and its Supporting Information files.

**Funding:** This work was supported by the National Sci-Tech Key Project (2018ZX10713-003-002,

## Abstract

The increasing prevalence of carbapenem-resistant *Acinetobacter baumannii* (CRAB) caused nosocomial infections generate significant comorbidity and can cause death among patients. Current treatment options are limited. These infections pose great difficulties for infection control and clinical treatment. To identify the antimicrobial resistance, carbapenemases and genetic relatedness of *Acinetobacter baumannii* isolates from cerebrospinal fluid (CSF) and blood, a total of 50 nonrepetitive CSF isolates and 44 blood isolates were collected. The resistance phenotypes were determined, and polymerase chain reaction (PCR) was performed to examine the mechanisms of carbapenem resistance. Finally, multilocus sequence typing (MLST) was conducted to determine the genetic relatedness of these isolates. It was observed that 88 of the 94 collected isolates were resistant to imipenem or meropenem. Among them, the $bla_{OXA-23}$ gene was the most prevalent carbapenemase gene, with an observed detection rate of 91.5% (86/94), followed by the $bla_{OXA-24}$ gene with a 2.1% detection rate (2/94). Among all carbapenem-resistant *Acinetobacter baumannii* (CRAB) observations, isolates with the $bla_{OXA-23}$ gene were resistant to both imipenem and meropenem. Interestingly, isolates positive for the $bla_{OXA-24}$ gene but negative for the $bla_{OXA-23}$ gene showed an imipenem-sensitive but meropenem-resistant phenotype. The MLST analysis identified 21 different sequence types (STs), with ST195, ST540 and ST208 most frequently detected (25.5%, 12.8% and 11.7%, respectively). 80 of the 94 isolates (85.1%) were clustered into CC92 which showed a carbapenem resistance phenotype (except AB13). Five novel STs were detected, and most of them belong to CRAB. In conclusion, these findings provide additional observations and epidemiological data of CSF and blood *A. baumannii* strains, which may improve future infection-control measures and aid in potential clinical treatments in hospitals and other clinical settings.

2018ZX10713-001-002). The funders had no role in study design, data collection and analysis, decision to publish, or preparation of the manuscript.

**Competing interests:** The authors have declared that no competing interests exist.

## Introduction

*Acinetobacter baumannii* is a non-fermentative, Gram-negative opportunistic pathogen that often causes disease among immunocompromised patients [1]. In recent years, *A. baumannii* has become an important bacterium to identify when treating and controlling infectious diseases because of its remarkable ability to evolve and develop extensive drug resistance to many antibiotics [2]. *A. baumannii* is often found in hospitals and causes a variety of nosocomial infections and iatrogenic diseases, including bloodstream infections, urinary tract infections, meningitis and wound infections [3]. Among them, microbiological results identifying bacteria in cerebrospinal fluid (CSF) and blood [4, 5] are referred to as critical values because these laboratory values may indicate a possibly urgent and life-threatening situation for patients in which treatment protocols indicate a need for immediate therapy. Therefore, the presence of *A. baumannii* strains in CSF and blood is an essential critical value to identify for practitioners, because this can pose great difficulties for clinical treatment options.

Carbapenems are considered to be the most effective antibiotics against many multidrug-resistant bacteria [6]. However, the increase in the number of carbapenem-resistant *Acinetobacter baumannii* (CRAB) isolates has recently become a global concern. CHINET surveillance data have shown that from 2005 to 2018, the resistance of *A. baumannii* to imipenem and meropenem has increased by approximately two-fold [7, 8]. The production of carbapenemase is one of the most common and important mechanisms for *A. baumannii* resistance to carbapenems. OXA-type carbapenemase (mainly OXA-23, OXA-24, OXA-48, OXA-51 and OXA-58) are most frequently found. Among them, OXA-23, OXA-24, OXA-48 and OXA-58 are acquired carbapenemases, whereas OXA-51 is intrinsic to *A. baumannii* strains [9–11]. In addition, because the related coding genes are located in transferable genetic elements and can spread among *A. baumannii* and even into other bacteria [12, 13], New Delhi metallo-β-lactamase (NDM) and *Klebsiella pneumoniae* carbapenemase (KPC) producers have also shown significant importance for worldwide prevalence [14, 15].

Multilocus sequence typing (MLST) has been widely used to determine genetic relatedness and for molecular epidemiological studies of *A. baumannii* [16, 17]. Previous molecular epidemiological studies have shown that CC92 is highly prevalent throughout China, and it is the clonal complex (CC) with the widest global distribution [18, 19].

In the present study, the antimicrobial resistance, carbapenemases, and genetic relatedness of 94 *A. baumannii* isolates obtained from CSF and blood at the First Affiliated Hospital of Shandong First Medical University (Shandong, China) was investigated. The aim of this study was to investigate the antimicrobial resistance, carbapenemases, and genetic relatedness of the *A. baumannii* isolates obtained from CSF and blood in a hospital in Shandong, China.

## Materials and methods

### Ethical statement

This study focused only on bacteria and did not include any human materials or patient information. The Ethics Committee of the First Affiliated Hospital of Shandong First Medical University exempted this study from review and the Review Board also waived the requirement for informed consent.

### Bacterial isolates

A total of 94 nonrepetitive *A. baumannii* isolates were obtained from CSF (n = 50) or blood (n = 44) samples of patients at the First Affiliated Hospital of Shandong First Medical University (Shandong, China). The samples used in this study were obtained from 2014 to 2019.

Among the CSF isolates, one isolate was collected in 2014, four in 2015, 17 in 2016, 15 in 2017, five in 2018, and eight in 2019. Forty-one of these isolates were obtained from neurosurgery, six isolates from the ICU, and three from other wards. Regarding the *A. baumannii* isolates samples obtained from blood, 12 were collected in 2016, 13 in 2017, 11 in 2018, and eight in 2019. Twenty-eight blood isolates were obtained from the ICU, six blood isolates from neurosurgery, four from hematopathology, and six from other hospital wards. The CSF or blood specimens were collected from patients with suspected bacterial meningitis or blood stream infections (BSIs) according to CDC criteria [20]. All isolates were identified using MALDI-TOF MS (Bruker, Germany) and further verified by polymerase chain reaction (PCR) products of 16S rDNA sequencing [21]. PCR products were sequenced by Tsingke BioTech Co., Ltd., followed by sequence alignment via the NCBI database.

## Antimicrobial susceptibility tests

All *A. baumannii* strains were tested for susceptibilities to 14 antibiotics, including ticarcillin/clavulanic acid, piperacillin/tazobactam, ceftazidime, cefoperazone/sulbactam, cefepime, imipenem, meropenem, tobramycin, ciprofloxacin, levofloxacin, minocycline, tigecycline, colistin and trimethoprim/sulfamethoxazole, using a Vitek-2 compact system (bioMérieux, Marcy, France) with AST-N-335 cards. The results were evaluated according to the Clinical and Laboratory Standards Institute (CLSI) criteria, except for tigecycline, in which case the results were evaluated from the adapted United States Food and Drug Administration breakpoints.

## PCR experiments

PCR assays were conducted using conventional PCR amplification. The target genes included the $bla_{OXA-51}$, $bla_{OXA-23}$, $bla_{OXA-24}$, $bla_{OXA-58}$, $bla_{OXA-48}$, $bla_{NDM-1}$, and $bla_{KPC}$ genes. Table 1 shows the sequences used for primer design and the annealing temperatures. Positive amplicons were randomly selected for sequencing to verify the amplicons sequences.

## Multilocus Sequence Typing (MLST)

MLST analyses were performed using the Oxford scheme [16]. Amplification reactions were carried out as described previously [19, 26]. The sequence types (STs) and allelic profiles were

**Table 1. Primers used with their respective annealing temperatures.**

| Primer | Sequence | Amplicon length | Annealing temp | Ref |
|---|---|---|---|---|
| $bla_{OXA-51}$-F | 5'-ATGAACATTAAAGCACTC-3' | 353 bp | 46°C | [22] |
| $bla_{OXA-51}$-R | 5'-CTATAAAATACCTAATTGTTC-3' | | | |
| $bla_{OXA-23}$-F | 5'-GATCGGATTGGAGAACCAGA-3' | 501 bp | 53°C | [22] |
| $bla_{OXA-23}$-R | 5'-ATTTCTGACCGCATTTCCAT-3' | | | |
| $bla_{OXA-24}$-F | 5'-GGTTAGTTGGCCCCCTTAAA-3' | 246 bp | 53°C | [22] |
| $bla_{OXA-24}$-R | 5'-AGTTGAGCGAAAAGGGGATT-3' | | | |
| $bla_{OXA-58}$-F | 5'-AAGTATTGGGGCTTGTGCTG-3' | 599 bp | 53°C | [22] |
| $bla_{OXA-58}$-R | 5'-CCCCTCTGCGCTCTACATAC-3' | | | |
| $bla_{KPC}$-F | 5'-GCTCAGGCGCAACTGTAAGT-3' | 823 bp | 55°C | [23] |
| $bla_{KPC}$-R | 5'-GTCCAGACGGAACGTGGTAT-3' | | | |
| $bla_{NDM-1}$-F | 5'-TCTCGACATGCCGGGTTTCGG-3' | 475 bp | 55°C | [24] |
| $bla_{NDM-1}$-R | 5'-ACCGAGATTGCCGAGCGACTT-3' | | | |
| $bla_{OXA-48}$-F | 5'-GCGTGGTTAAGGATGAACAC-3' | 438 bp | 52°C | [25] |
| $bla_{OXA-48}$-R | 5'-CATCAAGTTCAACCCAACCG-3' | | | |

analyzed using MLST database (https://pubmlst.org/abaumannii/info/primers_Oxford.shtml). The newly identified STs were submitted to the MLST database curator for approval, and an ST number was assigned. A minimum-spanning tree using the allelic difference between isolates of the seven housekeeping genes was constructed using BioNumerics (Applied Math).

### Statistical analysis

SPSS 21.0 (SPSS Inc., Chicago, IL, USA) was used for data analysis. The chi-square tests were performed to compare the differences of resistance rates and the distributions of carbapenem-resistance genes between CSF isolates and blood isolates. A two-sided $p < 0.05$ was considered to be statistically significant.

## Results

### Antibiotic susceptibilities

The antimicrobial susceptibility profiles of 94 *A. baumannii* strains are shown in Table 2, Fig 1, and the S1 Table. In general, more than 90% of the *A. baumannii* isolates were resistant to ticarcillin/clavulanic acid, piperacillin/tazobactam, ceftazidime, and ciprofloxacin. Furthermore, 88 of 94 *A. baumannii* isolates were CRAB isolates. Among them, 86 isolates were resistant to both imipenem and meropenem. Two isolates showed a meropenem-resistant but imipenem-sensitive phenotype. The changes in resistance rates of imipenem were observed to decrease from 89.7% in 2016 to 89.3% in 2017, and then increased to 93.8% in 2018 and 2019. For meropenem, the resistance rates were observed to increase from 89.7% in 2016 to 96.4% in 2017, and then decreased to 93.8% in 2018 and 2019. Most isolates (over 89%) were found to be resistant to these six antibiotics each year from 2016 to 2019 (Fig 1). Because only one strain was isolated in 2014 and only four strains in 2015, we omitted these results from our analysis. A total of 77.7% of the isolates were resistant to tobramycin, 75.5% were resistant to levofloxacin, 59.6% were resistant to trimethoprim/sulfamethoxazole, 52.1% were resistant to cefepime and 45.7% were resistant to cefoperazone/sulbactam. However, the resistance rates for levofloxacin, trimethoprim/sulfamethoxazole, and cefoperazone/sulbactam were observed to

**Table 2. Resistance rates of *A. baumannii* isolates obtained from CSF and blood.**

| Antimicrobial | Resistance rate (%) | | | P-value |
|---|---|---|---|---|
| | Overall | CSF | Blood | |
| | (n = 94) | (n = 50) | (n = 44) | |
| Ticarcillin/clavulanic acid | 87(92.6) | 47(94.0) | 40(90.9) | 0.860 |
| Piperacillin/tazobactam | 87(92.6) | 47(94.0) | 40(90.9) | 0.860 |
| Ceftazidime | 87(92.6) | 47(94.0) | 40(90.9) | 0.860 |
| Cefoperazone/sulbactam | 43(45.7) | 19(38.0) | 24(54.5) | 0.108 |
| Cefepime | 49(52.1) | 28(56.0) | 21(47.7) | 0.432 |
| Imipenem | 86(91.5) | 46(92.0) | 40(90.9) | 1.000 |
| Meropenem | 88(93.6) | 47(94.0) | 41(93.2) | 1.000 |
| Tobramycin | 73(77.7) | 41(82.0) | 32(72.7) | 0.281 |
| Ciprofloxacin | 87(92.6) | 47(94.0) | 40(90.9) | 0.860 |
| Levofloxacin | 71(75.5) | 38(76.0) | 33(75.0) | 0.910 |
| Minocycline | 18(19.1) | 6(12.0) | 12(27.3) | 0.060 |
| Tigecycline | 7(7.4) | 3(6.0) | 4(9.1) | 0.860 |
| Colistin | 0(0) | 0(0) | 0(0) | - |
| Trimethoprim/sulfamethoxazole | 56(59.6) | 34(68.0) | 22(50.0) | 0.076 |

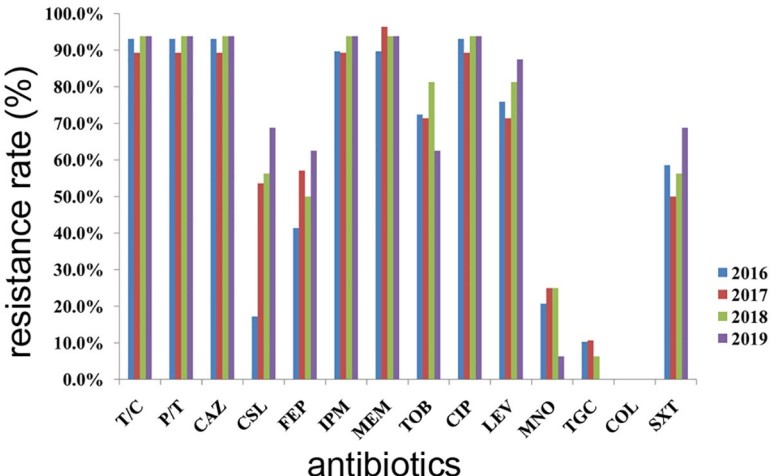

**Fig 1. Antibiotic resistance rates from 2016 to 2019.** T/C: ticarcillin/clavulanic acid; P/T: piperacillin/tazobactam; CAZ: ceftazidime; CSL: cefoperazone/sulbactam; FEP: cefepime; IPM: imipenem; MEM: meropenem; TOB: tobramycin; CIP: ciprofloxacin; LEV: levofloxacin; MNO: minocycline; TGC: tigecycline; COL: colistin; SXT: sulfamethoxazole/trimethoprim.

increase year by year from 2017 to 2019. In contrast, the resistance rates for minocycline and tigecycline were only 19.1% and 7.4%, respectively, which decreased year by year from 2017 to 2019. All isolates were sensitive to colistin. No statistically significant differences were observed between CSF and blood isolates among all antibiotics tested in this study.

## Distribution of carbapenem-resistance genes

All isolates were screened for the presence of carbapenem-resistance genes (Table 3). In the 94 *A. baumannii* isolates, all of them harbored the intrinsic $bla_{\text{OXA-51}}$ gene. In contrast, none of them had $bla_{\text{OXA-48}}$, $bla_{\text{OXA-58}}$, $bla_{\text{NDM-1}}$, or $bla_{\text{KPC}}$ genes. The $bla_{\text{OXA-23}}$ gene was the most prevalent carbapenemase gene, with a 91.5% (86/94) detection rate, followed by the $bla_{\text{OXA-24}}$ gene in only two (2.1%) isolates. No statistically significant differences were observed between CSF and blood isolates for all detected carbapenem-resistance genes.

In this study, all of the isolates with $bla_{\text{OXA-23}}$ or $bla_{\text{OXA-24}}$ genes were carbapenem- resistant. Among them, isolates with the $bla_{\text{OXA-23}}$ gene were resistant to both imipenem and meropenem. However, isolates positive for the $bla_{\text{OXA-24}}$ gene but negative for the $bla_{\text{OXA-23}}$ gene showed an imipenem-sensitive but meropenem-resistant phenotype.

**Table 3. Prevalence of carbapenem-resistance genes.**

| Gene | No. of positive isolates (% of 94) | No. distributing in | | P-value |
|---|---|---|---|---|
| | | CSF (n = 50) | Blood (n = 44) | |
| | | n (%) | n (%) | |
| $bla_{\text{OXA-23}}$ | 86(91.5) | 46 (92.0) | 40 (90.9) | 1.000 |
| $bla_{\text{OXA-24}}$ | 2(2.1) | 1 (2.0) | 1 (2.3) | 1.000 |
| $bla_{\text{OXA-48}}$ | 0 | 0 (0) | 0 (0) | - |
| $bla_{\text{OXA-51}}$ | 94(100) | 50 (100) | 44 (100) | - |
| $bla_{\text{OXA-58}}$ | 0 | 0 (0) | 0 (0) | - |
| $bla_{\text{NDM-1}}$ | 0 | 0 (0) | 0 (0) | - |
| $bla_{\text{KPC}}$ | 0 | 0 (0) | 0 (0) | - |

**Table 4. Allelic profiles of the novel STs found in this study.**

| STs | *gltA* | *gyrB* | *gdhB* | *recA* | *cpn60* | *gpi* | *rpoD* |
|---|---|---|---|---|---|---|---|
| 1967 | 1 | 34 | 3 | 2 | 2 | 178 | 3 |
| 1968 | 1 | 3 | 3 | 2 | 2 | 113 | 3 |
| 1969 | 1 | 17 | 135 | 12 | 23 | 98 | 6 |
| 1970 | 1 | 3 | 3 | 2 | 2 | 160 | 4 |
| 1971 | 36 | 34 | 59 | 28 | 4 | 279 | 3 |

## MLST profiles

MLST analysis revealed a total of 21 different STs, including 16 existing STs and five novel STs. The novel STs were submitted for ST assignment and included ST1967, ST1968, ST1969, ST1970 and ST1971. The profiles of the newly identified ST types are listed in Table 4. Among them, 89.4% (84/94) were represented by 11 main STs (having ≥2 isolates), and the prevalent STs were ST195, ST540, and ST208, accounting for 25.5% (24/94), 12.8% (12/94), and 11.7% (11/94), respectively (Fig 2 and the S2 Table).

ST195 was also the dominant ST in samples obtained from years 2016, 2017 and 2019. However, in 2018, this ST did not appear, and the dominant ST was changed to ST540 and ST191. ST540 and ST208 isolates were found in every year from 2016 to 2019, but the ratio changed year by year (Fig 3). Although the ST191 strain was not found in CSF or blood from 2014 to 2016, it emerged in 2017 and even rose to 25% in 2018. For other minor ST types, the

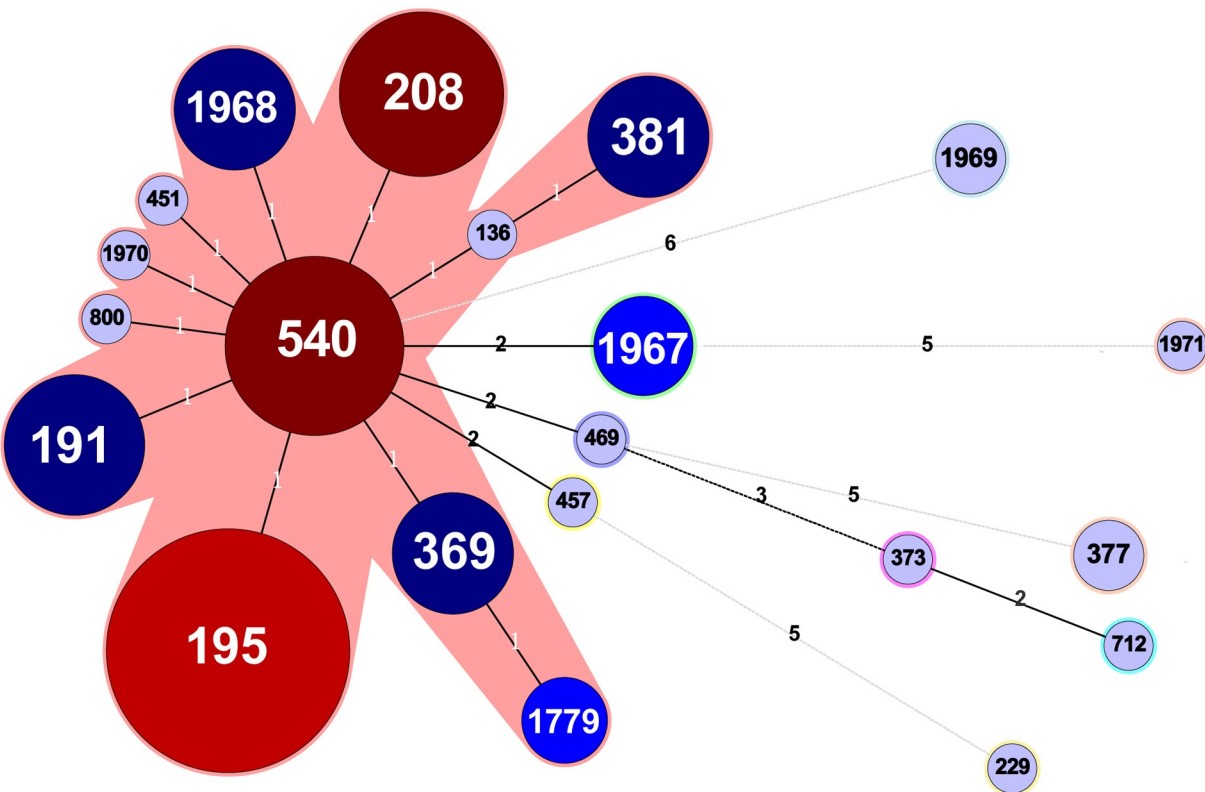

**Fig 2. Minimum-spanning tree of 94 *Acinetobacter baumannii* isolates from CSF and blood based on MLST.** Each ST is represented by a circle sized in proportion to the number of isolates represented by that ST, isolates within the pink shaded area are the CC92 isolates, and the number of allelic differences between STs is indicated on the branches. The detailed MLST profiles can be seen in the S2 Table.

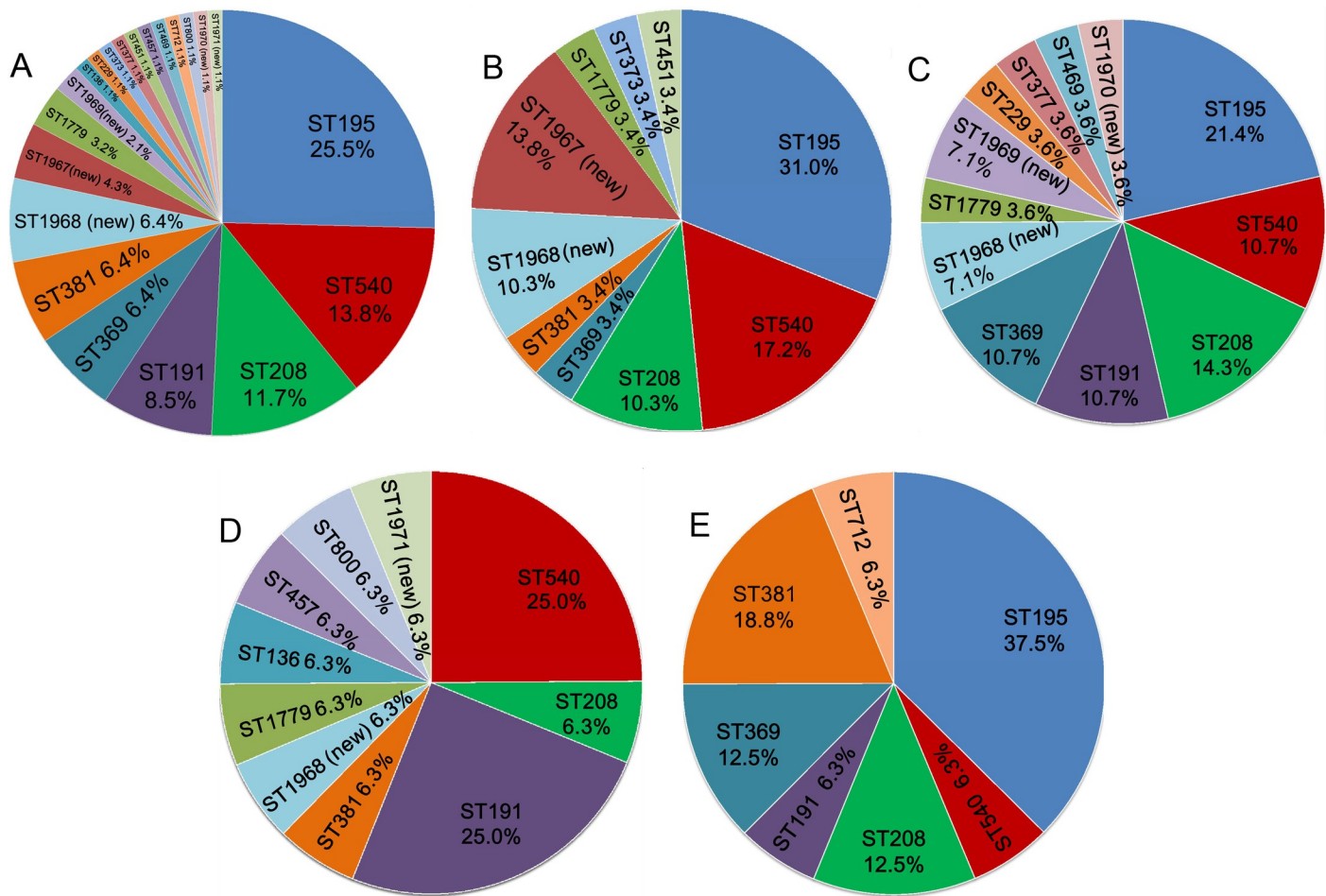

**Fig 3. MLST population analysis over the different years.** (A) ST in all 94 isolates. (B) ST distribution in 2016. (C) ST distribution in 2017. (D) ST distribution in 2018. (E) ST distribution in 2019.

distributions were varied across different years. For instance, the ratio of ST369 isolates increased from 3.4% in 2016 to 10.7% in 2017. In 2018, this type of strain was not observed but the ratio rose to 12.5% in 2019. Because only one strain was isolated in 2014 and four strains in 2015, we omitted these results from our analysis.

Twelve STs representing 85.1% (80/94) of the isolates were clustered into CC92, with up to 18 different allelic profiles and 56 different isolates being represented. In addition, 9 individual STs accounted for 14 isolates. The detailed MLST profiles are presented in the S2 Table.

## Molecular epidemiological characteristics

All of the isolates grouped into CC92 were carbapenem-resistant *A. baumannii* isolates, except for one isolate (AB13). These isolates were also not sensitive to ticarcillin/clavulanic acid, piperacillin/tazobactam, ceftazidime, cefepime, ciprofloxacin,or levofloxacin but had variable susceptibilities to cefoperazone/sulbactam, tobramycin, minocycline, tigecycline and trimethoprim/sulfamethoxazole. In contrast, all of the carbapenem-sensitive *Acinetobacter baumannii* (CSAB) isolates belonged to individual STs, except for one isolate (AB13). These isolates were also sensitive to the other 12 antibiotics tested in this study.

Five novel STs were identified in this study. All four ST1967 isolates were obtained from CSF and were non-sensitive to any β-lactam antibiotics or quinolones. For ST 1968 isolates, five were non-sensitive to 13 antibiotics (except colistin) tested in this study, but the other isolate (AB13) was sensitive to tobramycin, trimethoprim/sulfamethoxazole and colistin. The ST1970 isolate was non-sensitive to 12 antibiotics (except tigecycline and colistin) tested in this study, whereas the ST1971 isolate was sensitive to all antibiotics. ST1969 contained two isolates (AB29 and AB70). Both of these isolates possessed the $bla_{OXA-24}$ gene and were resistant to meropenem but sensitive to imipenem.

The $bla_{OXA-23}$ gene was more prevalent than the $bla_{OXA-24}$ gene, but the detection rates were altered among different STs. All isolates in CC92 (containing 12 STs) carried the $bla_{OXA-23}$ gene (except one isolate in ST208). In contrast, the $bla_{OXA-23}$ gene was detected in only 50% (7/14) of isolates of individual STs. None of the isolates in ST373, ST377, ST712, ST1969, or ST1971 possessed this gene.

## Discussion

CSF and blood infection of *A. baumannii* (especially for CRAB) may be life-threatening and present great obstacles for effective clinical treatments [4, 5]. Our present study offers insights into the molecular characterization and antibiotic resistance of *A. baumannii* from CSF and blood.

CHINET surveillance data have shown that, in China, from 2005 to 2018, the resistance rate of *A. baumannii* for imipenem and meropenem increased from 32.9% to 71.7% and from 41.3% to 78.1%, respectively [7, 8]. Compared to those of the surveillance data, the resistance rates of *A. baumannii* for imipenem and meropenem in our present study were 91.5% and 93.6%, respectively, which were more than twofold higher than the surveillance data in 2005 and more than 10% higher than those in 2018. For CSF and blood infections, the recommended doses of antibiotics are usually higher and require longer courses of treatment than for superficial infections. This may have contributed to the higher drug-resistance rates in our study than those reported in CHINET surveillance data, which also include superficial infections. In addition, the CHINET surveillance data [7, 8, 27, 28] have demonstrated that the resistance rate of *A. baumannii* to meropenem is slightly higher than that to imipenem. This is similar to our present observations and results. In fact, our results also support the view that imipenem is more bactericidal [29] and has a higher Time>MIC value [30] than that of meropenem against *A. baumannii*. In contrast, the resistance rate to tigecycline was only 7.4%, and all isolates showed a colistin-sensitive phenotype. The resistance rates of these two drugs were far below those of carbapenem and the other antibiotics tested in the present study. In this hospital, the frequencies of the usages of tigecycline and colistin were 10 times and 492 times lower than those of carbapenems. This may have contributed to the low resistance rates of these two antibiotics. Additionally, because of these low resistance rates, tigecycline may serve as a preferred therapeutic agent to control CRAB infections in this hospital (S3 Table). Although colistin also had a low resistance rate in our present study, fewer patients were treated by this antibiotic, as its side effects of nephrotoxicity and neurotoxicity [31].

To investigate the mechanism of carbapenem resistance, carbapenemase-encoding genes were assessed in the present study. Our results showed that the $bla_{OXA-23}$ gene was present in most CRAB isolates but was absent in all CSAB isolates, which suggests that $bla_{OXA-23}$ represents the primary mechanism underlying carbapenem resistance of CRAB isolated from CSF and blood. The $bla_{OXA-23}$ gene was also found to be the most important mechanism underlying carbapenem resistance in CRAB in China [19, 26] and some other countries [32–34]. In addition, in our present study, the $bla_{OXA-24}$ gene was found to be another mechanism behind

carbapenem resistance, as $bla_{OXA-24}$-positive but $bla_{OXA-23}$-negative isolates in this study showed meropenem-resistance but imipenem-sensitive characteristics. $Bla_{OXA-24}$-positive *A. baumannii* strains have been reported in many countries [11, 35, 36], especially in Spain, where $bla_{OXA-24}$ has been shown to be the most prevalent gene [11]. Interestingly, even though most $bla_{OXA-24}$-positive isolates have been reported to be resistant to both imipenem and meropenem, in our experiment, $bla_{OXA-24}$-positive strains showed an imipenem-sensitive but meropenem-resistant phenotype. Some molecular biological mechanisms have been reported in many Gram-negative bacteria to explain this imipenem-sensitive but meropenem-resistant phenomenon. For example, the transmission of $bla_{IMP-6}$ and $bla_{CTX-M-2}$ plasmids [37], as well as the absence of OmpK35 and the frame shift mutation in OmpK36 [38], has been shown to be important mechanisms for imipenem-sensitive but meropenem-resistant *Klebsiella pneumoniae* (ISMRKP) strains. In *Pseudomonas aeruginosa*, however, substrate specificities of efflux pumps lead to different drug resistance characteristics. As a specific substrate, meropenem can be extruded by many efflux pumps, but imipenem is not affected by these efflux systems [39]. As a result, some imipenem-sensitive but meropenem-resistant *Pseudomonas aeruginosa* strains were detected. However, few studies have examined this mechanism against *A. baumannii*. Both of these strains were found to belong to a novel ST (ST1969) in our present study. Hence, this molecular mechanism for *A. baumannii* strains requires further investigation.

ST540, ST195, and ST208 were three major STs for *A. baumannii* isolated from CSF and blood in our present study. Among them, ST195 and ST208 are two dominant STs currently found in China [40–42]. In the present study, ST 195 was also the dominant ST in 2016, 2017, and 2019. However, no ST195 isolates were found in 2018. Although ST540 is not the main ST observed in China, our data shown that ST540 was not only one of the three common STs but also the predicted founder of the CC for *A. baumannii* isolated from blood and CSF. ST208 isolates were detected every year from 2016 to 2019, but the ratio changed year by year. High detection rates of CC92 *A. baumannii* isolates in CSF and blood, as well as a high correlation between CC92 and carbapenem-resistance characteristics, were found in the present study. This is consistent with some other studies that have suggested that CC92 is the largest and most geographically diverse CC, which is widespread in many countries [40], including China [26]. As CC92 is a widespread variant that has advantages in acquiring resistance determinants and surviving in the nosocomial environment [19], this may have contributed to the high correlation between CC92 and carbapenem resistance characteristics in our present study.

We also observed a total of five novel STs. Among them, two novel STs were classified into CC92 and others were individual STs. As 13 of the 14 isolates in the five novel STs were identified as CRAB, close attention should be paid toward these novel STs to identify and further limit both transmission and outbreaks.

## Conclusions

In summary, we examined and described the molecular characterization and antibiotic resistance of *A. baumannii* from CSF and blood in a hospital in Shandong, China. A high level of carbapenem resistance was detected. The majority of the isolates were carbapenem-resistant *Acinetobacter baumannii* (CRAB), which carried the $bla_{OXA-23}$ carbapenemase gene and belonged to MLST CC92. Five novel STs were detected, and most of them were CRAB, some of which belonged to CC92. Collectively, our findings offer new epidemiological data of CSF and blood *A. baumannii* strains, which may help to improve infection control measures and clinical treatments in hospitals.

## Supporting information

**S1 Table. Minimum Inhibitory Concentrations (MICs) of the tested antimicrobial agents.**
T/C: ticarcillin/clavulanic acid; P/T: piperacillin/tazobactam; CAZ: ceftazidime; CSL: cefopera-
zone/sulbactam; FEP: cefepime; IPM: imipenem; MEM: meropenem; TOB: tobramycin; CIP:
ciprofloxacin; LEV: levofloxacin; MNO: minocycline; TGC: tigecycline; COL: colistin; SXT:
sulfamethoxazole/trimethoprim.
(DOCX)

**S2 Table. Allelic profiles of the *A. baumannii* strains used in this study.**
(DOCX)

**S3 Table. Antibiotics used for the CRAB treatments.**
(DOCX)

## Acknowledgments

We thank Duochun Wang and Tao Xiao for some helpful comments on our manuscript.

## Author Contributions

**Conceptualization:** Jiazheng Wang.

**Data curation:** Xiaohong Shi, Hong Wang.

**Formal analysis:** Jiazheng Wang.

**Project administration:** Ran Duan, Shuai Qin, Dongyue Lv, Yufeng Fan, Zhenzhou Huang,
Jiazheng Wang.

**Resources:** Hong Wang.

**Writing – original draft:** Xiaohong Shi, Jiazheng Wang.

**Writing – review & editing:** Xin Wang, Huaiqi Jing, Kyle Stirling, Lei Zhang.

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
