## [Decision Letter · Decision Letter 0]

13 Nov 2020

PONE-D-20-30754

Molecular characterization and antibiotic resistance of Acinetobacter baumannii in cerebrospinal fluid and blood

PLOS ONE

Dear Dr. Wang,

Thank you for submitting your manuscript to PLOS ONE. After careful consideration, we feel that it has merit but does not fully meet PLOS ONE’s publication criteria as it currently stands. Therefore, we invite you to submit a revised version of the manuscript that addresses the points raised during the review process.

The manuscript requires substantial revision from both language and scientific content perspective. Please see the comments made directly on the manuscript and additional referee comments below.

We look forward to receiving your revised manuscript.

Kind regards,

Iddya Karunasagar

Academic Editor

PLOS ONE

Additional Editor Comments (if provided):

Two reviewers have commented on the manuscript and raised very pertinent points. Some comments have been made directly on the manuscript. Please address these comments point by point.

Journal Requirements:

2) Thank you for stating the following financial disclosure:

 [There are no financial conflicts of interest to disclose].

Reviewers' comments:

Reviewer's Responses to Questions

**Comments to the Author**

1. Is the manuscript technically sound, and do the data support the conclusions?

Reviewer #1: Yes

Reviewer #2: Yes

2. Has the statistical analysis been performed appropriately and rigorously? 

Reviewer #1: N/A

Reviewer #2: Yes

3. Have the authors made all data underlying the findings in their manuscript fully available?

Reviewer #1: Yes

Reviewer #2: Yes

4. Is the manuscript presented in an intelligible fashion and written in standard English?

Reviewer #1: No

Reviewer #2: Yes

5. Review Comments to the Author

Reviewer #1: The study by Wang et al analysed A. baumannii isolates obtained from human CSF and blood samples. The methods used included PCR to detect bla-oxa carbapenemase genes, MLST and AST.

The MS needs major language revision, and the figures, especially fig. 3 need to be modified, i.e. the same colour scheme should be applied to the same ST.

The MS would benefit enormously if it were shortened to a letter or note or something alike. The methods and analyses used are not new and do not have to be elaborated. Moreover, the results are also not unexpected, i.e. presence of IC II (CC92) in China, presence of highly resistant A.b strains, further clonal expansion.

Reviewer #2: This study gives an insight into epidemiology and antibiotic resistance pattern and mechanism of carbapenem of A. baumannii strains isolated from CSF and Blood. The study includes 50 isolates from CSF - are all these isolates were true pathogens the contamination has been ruled out to be clarified.

Comments are included in the manuscript which can be clarified. Reasons for high resistance level to Carbapenems and low rate of resistance to Tigicycline and levofloxacin to be discussed

The manuscript needs correction for English as suggested in the comments in the manuscript.

Wheather the treatment and outcome of the patient has been looked into if so which are the drugs used for the treatment of these infections can be mentioned.

6. PLOS authors have the option to publish the peer review history of their article (what does this mean?). If published, this will include your full peer review and any attached files.

Reviewer #1: No

Reviewer #2: No

---

## [Author Response · Author response to Decision Letter 0]

24 Dec 2020

We appreciate the interest that the editors and reviewers have expressed in our manuscript and the constructive comments they have given. Those comments are all valuable and very helpful for revising and improving our paper, as well as the important guiding significance to our researches. We have carefully revised the manuscript according to the insightful comments and provided point-by-point responses; including how and where the text was modified. Please see the file labeled 'Response to Reviewers'and the revised manuscript.

---

## [Decision Letter · Decision Letter 1]

5 Jan 2021

PONE-D-20-30754R1

Molecular Characterization and Antibiotic Resistance of Acinetobacter baumannii in Cerebrospinal Fluid and Blood

PLOS ONE

Dear Dr. Wang,

Thank you for submitting your manuscript to PLOS ONE. After careful consideration, we feel that it has merit but does not fully meet PLOS ONE’s publication criteria as it currently stands. Therefore, we invite you to submit a revised version of the manuscript that addresses the points raised during the review process.

The manuscript language needs significant improvement

We look forward to receiving your revised manuscript.

Kind regards,

Iddya Karunasagar

Academic Editor

PLOS ONE

Additional Editor Comments (if provided):

The authors have addressed technical comments but the language still requires significant improvement.

Reviewers' comments:

Reviewer's Responses to Questions

**Comments to the Author**

1. If the authors have adequately addressed your comments raised in a previous round of review and you feel that this manuscript is now acceptable for publication, you may indicate that here to bypass the “Comments to the Author” section, enter your conflict of interest statement in the “Confidential to Editor” section, and submit your "Accept" recommendation.

Reviewer #1: (No Response)

Reviewer #2: All comments have been addressed

2. Is the manuscript technically sound, and do the data support the conclusions?

Reviewer #1: Yes

Reviewer #2: Yes

3. Has the statistical analysis been performed appropriately and rigorously? 

Reviewer #1: Yes

Reviewer #2: Yes

4. Have the authors made all data underlying the findings in their manuscript fully available?

Reviewer #1: Yes

Reviewer #2: No

5. Is the manuscript presented in an intelligible fashion and written in standard English?

Reviewer #1: No

Reviewer #2: No

6. Review Comments to the Author

Reviewer #1: The authors have done a thorough investigation on clinical A. baumannii isolates obtained from CSF and blood at the Shandong Hospital and the results are certainly important in the Chinese and global setting. However, the MS still needs a rigorous language revision to improve readability before it can be considered for publication in PLOS one or other journals.

Reviewer #2: All the reviewer comments are addressed and the manuscript is revised accordingly. The manuscript and the data presented will add therapeutic applicability in the treatment of blood and CSF infections caused by Acinetobacter spp

7. PLOS authors have the option to publish the peer review history of their article (what does this mean?). If published, this will include your full peer review and any attached files.

Reviewer #1: No

Reviewer #2: No

---

## [Author Response · Author response to Decision Letter 1]

1 Feb 2021

Dear Editors and Reviewers:

We appreciate the interest that the editors and reviewers have expressed in our manuscript and the constructive comments they have given. We have re-scrutinized to improve the English by a language editing service, “LetPub” to edit the manuscript. We also added the information of the antibiotics used for the treatment in S3 table. We believe the manuscript has been greatly improved. Please see the revised manuscript.

Reviewer #1: Comments to the Author

The authors have done a thorough investigation on clinical A. baumannii isolates obtained from CSF and blood at the Shandong Hospital and the results are certainly important in the Chinese and global setting. However, the MS still needs a rigorous language revision to improve readability before it can be considered for publication in PLOS one or other journals.

Answer: Thanks for your comments and suggestion. We have re-scrutinized to improve the English by a language editing service, “LetPub” to edit the manuscript. We believe the manuscript has been greatly improved. Please see the revised manuscript. 

 

Reviewer #2: Comments to the Author

All the reviewer comments are addressed and the manuscript is revised accordingly. The manuscript and the data presented will add therapeutic applicability in the treatment of blood and CSF infections caused by Acinetobacter spp

Answer: We appreciate the reviewer’s comments and suggestion. We added the information of the antibiotics used for the treatment in S3 table and made language revision by using language editing service of “LetPub”. We believe the manuscript has been greatly improved. Please see the revised manuscript.

---

## [Editor Report · Decision Letter 2]

8 Feb 2021

Molecular Characterization and Antibiotic Resistance of Acinetobacter baumannii in Cerebrospinal Fluid and Blood

PONE-D-20-30754R2

Dear Dr. Wang,

We’re pleased to inform you that your manuscript has been judged scientifically suitable for publication and will be formally accepted for publication once it meets all outstanding technical requirements.

Kind regards,

Iddya Karunasagar

Academic Editor

PLOS ONE

Additional Editor Comments (optional):

All reviewer comments have been addressed.
---

## [Editor Report · Acceptance letter]

12 Feb 2021

PONE-D-20-30754R2 

Molecular Characterization and Antibiotic Resistance of *Acinetobacter baumannii* in Cerebrospinal Fluid and Blood 

Dear Dr. Wang:

I'm pleased to inform you that your manuscript has been deemed suitable for publication in PLOS ONE. Congratulations! Your manuscript is now with our production department. 

Kind regards, 

on behalf of

Dr. Iddya Karunasagar 

Academic Editor

PLOS ONE